# Features of the Hydrocarbon Distribution in the Bottom Sediments of the Norwegian and Barents Seas

**Inna A. Nemirovskaya * and Anastasia V. Khramtsova**

Shirshov Institute of Oceanology of Russian Academy of Sciences, 117997 Moscow, Russia;
asya-medvedeva95_16@mail.ru
* Correspondence: nemir44@mail.ru

**Abstract:** The results of the study of hydrocarbons (HCs): aliphatic (AHCs) and polycyclic aromatic hydrocarbons (PAHs) in bottom sediments (2019 and 2020, cruises 75 and 80 of the R/V Akademik Mstislav Keldysh) in the Norwegian-Barents Sea basin: Mohns Ridge, shelf Svalbard archipelago, Sturfiord, Medvezhinsky trench, central part of the Barents Sea, Novaya Zemlya shelf, Franz Victoria trough are presented. It has been established that the organo-geochemical background of the Holocene sediments was formed due to the flow of sedimentary material in the coastal regions of the Barents Sea on shipping routes. The anthropogenic input of HCs into bottom sediments leads to an increase in their content in the composition of $C_{org}$ (in the sandy sediments of the Kaninsky Bank at an AHC concentration up to 64 μg/g, when its proportion in the composition of $C_{org}$ reaches 11.7%). The endogenous influence on the of the Svalbard archipelago shelf in Sturfiord and in the Medvezhinsky Trench determines the specificity of local anomalies in the content and composition of HCs. This is reflected in the absence of a correlation between HCs and the grain size composition of sediments and $C_{org}$ content, as well as a change in hydrocarbon molecular markers. At the same time, the sedimentary section is enriched in light alkanes and naphthalene's that may be due to emission during point discharge of gas fluid from sedimentary rocks of the lower stratigraphic horizons and/or sipping migration.

**Keywords:** hydrocarbons (aliphatic and polycyclic aromatic hydrocarbons); organic matter; bottom sediments; alkanes; fluid flows; Norwegian Sea; Barents Sea





## 1. Introduction

The Norwegian-Barents Sea basin is one of the most promising areas for the development of shelf resources [1]. Anomalies of the OM components in distribution and of hydrocarbons (HCs) in their composition can be direct indicators of their origin [2–4].

Besides, the study of the composition, distribution and genesis of (HCs) in bottom sediments is necessary for subsequent geoecological control during exploration and mining [5–8].

OM and HCs of waters and bottom sediments usually show a complex composition [5,9]. These are autochthones and allochthones components with different origin. The former are syngenetic to the environment and consist of products of bio- and geochemical processes of OM transformation taking place in the water column during sedimentogenesis and at the beginning stages of burial in sediments [2,10]. The second epigenetic category of HCs is even more diverse. It includes products migrating from sedimentary strata, where their formation takes place during catagenesis and in the harsh conditions of metamorphism [11]. Furthermore, the HCs could contain anthropogenic components that enter the aquatic environment (especially in shallow waters) and bottom sediments with oil and oil products polluting water areas [5,7,12].

In different years, the research on the cruises of the R/V Akademik Mstislav Keldysh in the Norwegian and Barents Seas covered water and sediments in hydrothermal fields located within the Jan Mayenne axial volcanic uplift of the Mohns Ridge; in places of

outcrops of cold methane seeps on the continental margins of the Spitsbergen archipelago; the Medvezhinsky Trench, the Novaya Zemlya shelf, and craters in the central part of the Barents Sea were examined [13–16] (Figure 1).

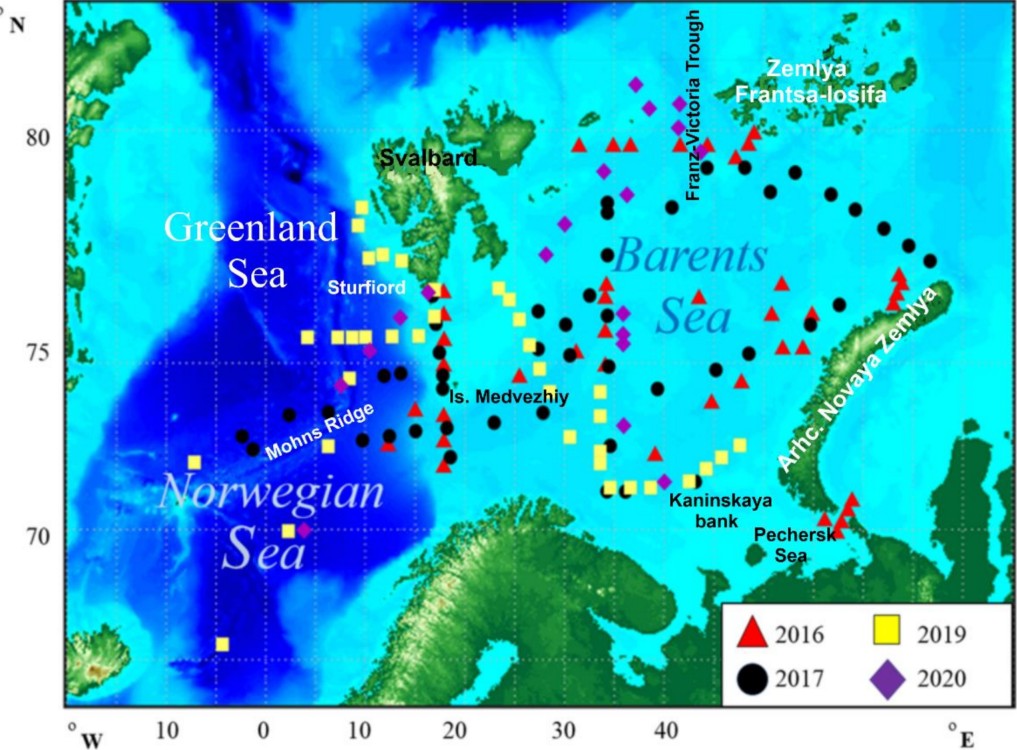

**Figure 1.** Scheme of stations in different years of research.

The purpose of our research is to obtain new data (2019 and 2020) on the spatial distribution and composition of aliphatic (AHCs) and polycyclic aromatic hydrocarbons (PAHs) in the bottom sediments of the Norwegian and Barents Seas, to establish contribution of HCs vertical migration to their total hydrocarbon pool in bottom sediments.

## 2. Methods of the Studies

The upper layer of bottom sediments was sampled with a Russian-made Okean-25 bottom grab, and undisturbed cores were sampled with a multicore, Mini Muc. K/MT410, KUM, Germany. To determine the moisture content of sediments, the samples was dried at 100 °C in a drying oven to constant weight. To determine Eh, a portable pH 3110 ionometer (WTW, Germany, Frankfurt) with selective electrodes was used—WTW Electrode Sen Tix ORP.

All solvents were of high purity grade. Methylene chloride was used to extract lipids from bottom sediments. The individual AHCs fractions were separated with hexane by means of column chromatography on silica gel. The concentrations of lipids and AHCs (before and after the chromatography, respectively) were determined by IR spectroscopy using an IR Affinity 1 instrument (Shimadzu, Japan, Kyoto). A mixture of isooctane, hexadecane, and benzene (37.5, 37.5, and 25 vol%, respectively) was used as a standard [5,8,17,18]. The sensitivity of the procedure amounted to 3 µg/mL of the extract [5].

The content and composition of PAHs were determined by the method of high performance liquid chromatography (HPLC) on a LC-20 Prominence liquid chromatograph (Shimadzu, Japan, Kyoto). An Envirosep PP column was used at 40 °C in a thermostat under gradient conditions (up to 50 to 90% in a volume of acetonitrile in water). A 1 cm$^3$/min flow rate of the eluent was used, and a RF 20A fluorescent detector with programmed wavelengths of absorption and excitation. The calculations were performed by means of LC Solution software. The equipment was standardized with individual

PAHs and their mixtures manufactured by Supelco Co (Sigma-Aldrich, Germany, Darmstadt). As a result, the key polyarenes recommended for studying the pollution of marine objects [12,19] were identified: naphthalene (Naph), 1-methylnaphthalene (1-MeNaph), 2-methylnaphthalene (2-MeNaph), acenaphthene (ACNF), fluorene (FL), phenanthrene (PHEN), anthracene (ANTR), fluoranthene (FLT), pyrene (PYR), benzo(a)anthracene (BaA), chrysene (CHR), benzo(e)pyrene (BeP), benzo(a)pyrene (BaP), benzo(b)fluoranthene (BbF), benzo(k)fluoranthene (BkF) dibenzo(a,h)anthracene (DbhA), indeno(1,2,3-c,d)pyrene (INP), and benzo(g,h,i)perylene (BPl).

The organic carbon in the samples of the SPM was determined by dry combustion with the TOC-L, (Shimadzu, Japan, Kyoto). The sensitivity amounted to 6 µg of carbon in a sample at a precision of 3–6%. The AHCs concentrations were converted into $C_{org}$ a factor of 0.86 [5,20].

## 3. Results

### 3.1. 2019 (The 75th Cruise of the R/V Akademik Mstislav Keldysh)

The sediments of the Jan-Mayen fault on the surface were represented by hydrothermal agglomerate silt with lumps of dark gray and bluish color. Red spots of ferruginization, rock fragments, shells, and pebbles were visible on the surface of the soft sediment. The AHC concentrations varied in the range of 5–51 µg/g (Supplementary Materials, Table S1), with the maximum value in the finely dispersed sediment of st. 6131 (Figure 2).

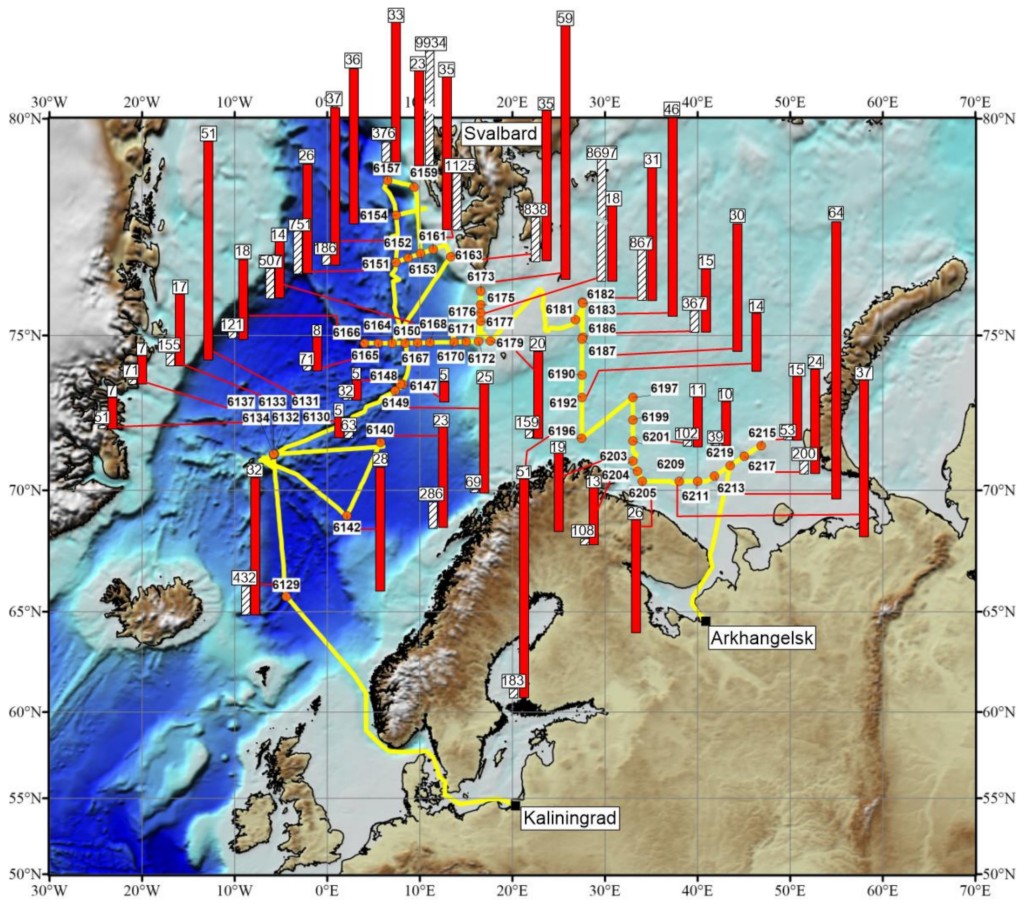

**Figure 2.** Distribution in the surface layer of bottom sediments of AHC concentrations (µg/g, values are shown above red columns) and PAHs (ng/g, values are above light columns). Squares 6129-6219 show the station numbers (the 75th cruise of the R/V Akademik Mstislav Keldysh).

The bottom sediments of the Barents Sea could be divided into three layers, which reflect three main stages of postglacial sedimentogenesis [21]. At the early stage, Holocene

BS formed, showing the predominance of pelite fraction formed due to fine products of glacier melting supplied to the basin. The second stage corresponds to the time of Atlantic climatic optimum with its maximal transgression level, active shore abrasion, and increasing role of sand-aleurite fraction. The third stage—the deposition of modern Late Holocene BS—falls on the subboreal–subatlantic period of basin regression, corresponding to an increase in the amount of pelite particles because of the stabilization of the rate of shore abrasion, supplying coarsegrained material.

In the Barents Sea, a high content of AHCs was found on the Eastern shelf of the Spitsbergen archipelago (51 µg/g, station 6196), and the maximum concentration was found in the southern part (64 µg/g, station 6213) on the North of the Kaninsky Bank (Figure 2). Light homologues slightly prevailed in the composition of n-alkanes of the surface layer of bottom sediments in most areas with the ratio L/H = $\sum(C_{12}–C_{24})/\sum(C_{25}–C_{37})$, averaged 1.22 (Table S2), which may indicate the intensity of autochthonous processes. The lowest values of this ratio were found on the Scandinavian shelf (stations 6203, 0.22) and in the northern part of the Pechora Sea (station 6217, 0.28), the maximum (station 6179, 2.06) was revealed in the eastern part of the latitudinal section in the Kveitola Trough.).

The area of the Mohns Ridge (station 6131), the Lofoten Basin (station 6142) and the area of the Knipovich Ridge (station 6154) shows microbial even alkanes domination in the low molecular weight area and a series of odd $C_{25}–C_{31}$ homologues prevails in the high molecular weight area (Figure 3). Low CPI values (1.40–2.06, average 1.64) and close concentrations of iso-compounds (pristane/phytane ratio—1.01 on average) may indicate a slight transformation of alkanes.

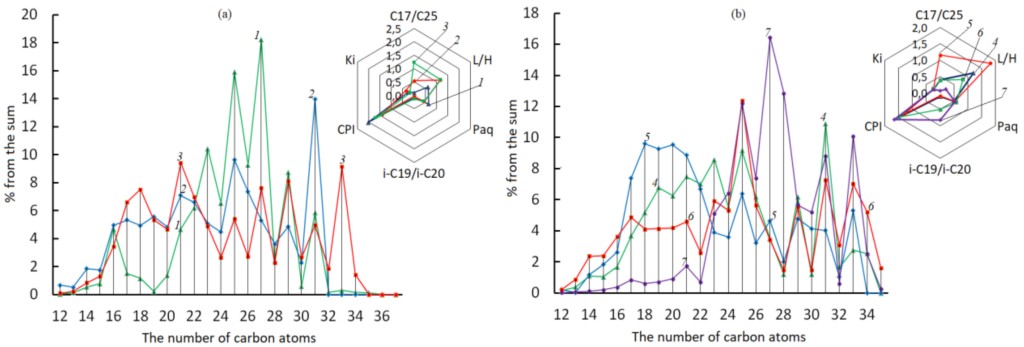

**Figure 3.** Composition of n-alkanes in the surface layer of bottom sediments at individual stations (75 cruise R/V Akademik Mstislav Keldysh, 2019)–(**a**), *1*–station 6131, *2*–station 6142, *3*–station 6154, (**b**), *4*–station 6190, *5*–station 6192, *6*–station 6213, *7*–station 6217. The inset shows the distribution of the main markers:. L/H = $\sum(C_{12}–C_{24})/\sum(C_{25}–C_{37})$; Paq = $(C_{23}+C_{25})/(C_{23}+C_{25}+C_{29}+C_{31})$; CPI = $\sum(odd)/\sum(even)$; $K_i = (i\text{-}C_{19}+i\text{-}C_{20})/(C_{17}+C_{18})$.

### 3.2. 2020 (The 80th Cruise of the R/V Akademik Mstislav Keldysh)

The range of AHCs concentrations in bottom sediments was significantly larger than in 2019: 3–186 µg/g (Figure 4). Nevertheless, in the area of the Mohns Ridge, a rather low content of both AHC (on average 14 µg/g) and $C_{org}$ was found. (average 0.44%). The sediments here are represented by sandy-silty-pelitic silt from dark boggy to almost black color with a small admixture of gravel and pebble material of volcanic origin (pyroclastic material). On the surface of the sediment, Fe-Mn crusts ranging in size from 1 to 10 cm thickness are noted.

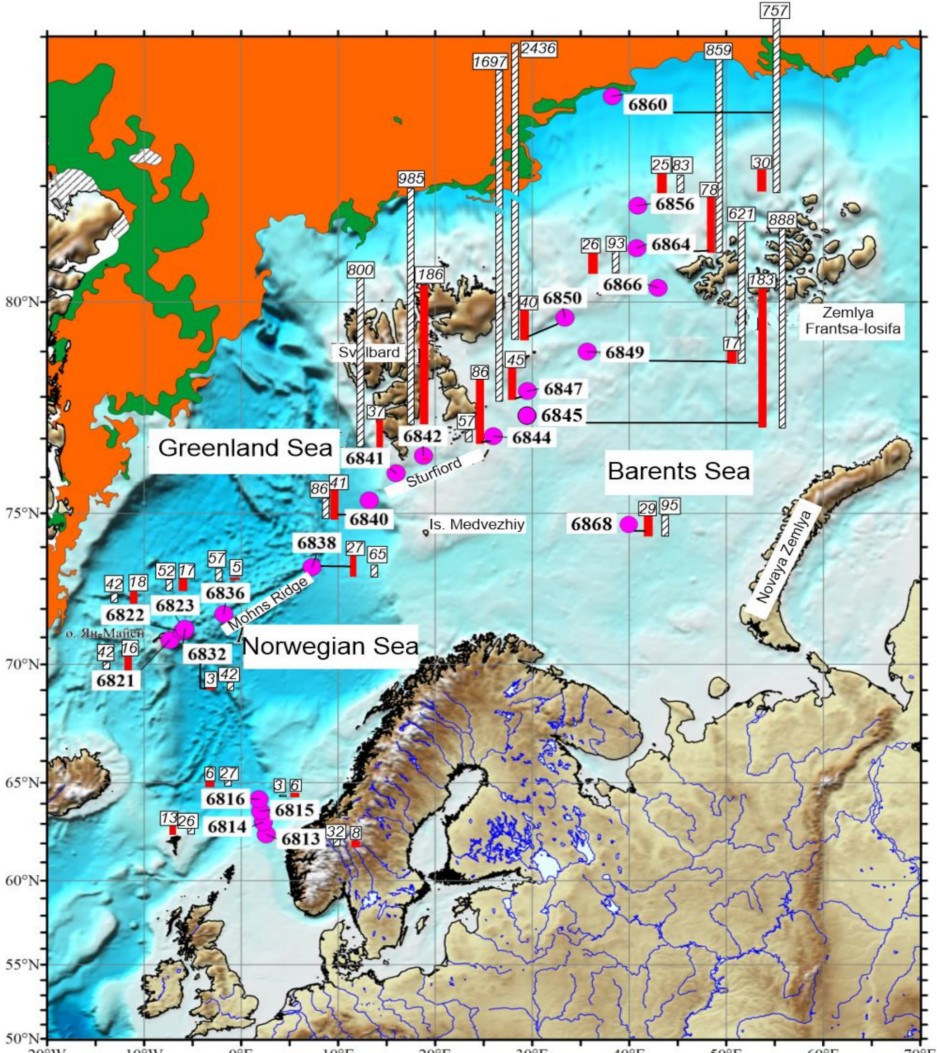

**Figure 4.** Distribution of AHC concentrations (at the top of the red columns, μg/g) and PAHs (at the top of the light columns, ng/g) in the surface layer of bottom sediments in the area of the 80th cruise of the R/V Akademik Mstislav Keldysh (2020); purple color shows the location of stations (station numbers in white squares).

The highest AHC concentrations in 2020 were found in Sturfiord (on average 90 μg/g, Table S1), with the maximum content at station 6842. Concentrations of AHCs were significantly lower on the eastern shelf of Svalbard (average 52 μg/g, Table S1).

Favorable ice conditions provided a unique opportunity to conduct research within the Franz Victoria Trench from depths of 3700 m (station 6860) to the shelf with a depth of 593 m (station 6864) and 403 m (station 6866), i.e., within the water area usually covered with ice. Here, the content of AHC (average 25 μg/g) in the surface layer of sediments was the lowest (Table S1).

The composition of alkanes in the surface layer of sediments was quite varied (Figure 5). On the Svalbard shelf, most samples were dominated by light homologues (stations 6844, 6845, L/H = 1.24–1.40) with low CPI values (1.23–1.69). On the other hand, in the Franz Victoria Trough (station 6860), the amount of high-molecular-weight alkanes increased (L/H = 0.53), but the CPI values (1.97) only slightly increased.

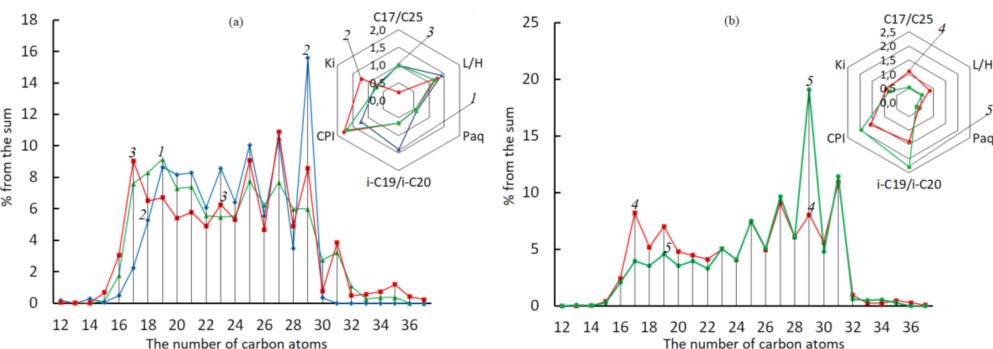

**Figure 5.** Composition of n-alkanes in the surface layer of bottom sediments at individual stations (80 cruise R/V Akademik Mstislav Keldysh, 2020) (**a**), *1*—station 6844, *2*—station 6845, *3*—station 6847, (**b**), *4*—station 6849, *5*—station. 6860. Inset: the distribution of markers in their composition.

The distribution of PAHs in surface sediments is mosaic, and their total content in 2019 varied in the range of 32–9934 ng/g, and in 2020—3–2430 ng/g (Figure 4). Such a wide concentration range is apparently due not only to different lithological-facial conditions of sedimentation, but also to the relative variability of the main geochemical parameters in the sedimentary strata.

The concentrations of PAHs found in the Mohns Ridge area averaged 50 ng/g were rather low. In Sturfiord, the PAH content at station 6842 (985 ng/g) with the maximum AHC content was also higher than at station 6841 (800 ng/g). However, in contrast to AHCs, the highest PAH concentrations are confined to the shelf of Svalbard. Therefore, there is no correlation between these hydrocarbon classes in the distribution of their concentrations in the surface layer: $r = 0.30$, $n = 21$.

In the northern part of the Barents Sea, in the sediments of the Franz Victoria Trench, their concentrations in the surface layer varied from 83 to 859 ng/g, with a maximum at station 6864.

In the bottom sediments thikness, the concentrations of both $C_{org}$ and HCs decreased with the depth of burial only at some stations only (Figure 6a). However, in most areas, there was an increase in their content in individual horizons. An example of such a distribution is the Sturfiord sedimentary stratum at station 6841 (Figure 6a). The sediment on the surface consisted of olive-brown silt-pelitic, which became darker with the depth of burial. When collecting the sediment, the smell of hydrogen sulfide was felt. In the sedimentary sequence (>2 cm), a large number of hydrotroilite smears and authigenic carbonate crusts were observed. The maximum concentration (218 µg/g) was established with a change in the redox potential in the mountains 6–7 cm (Eh = −80), where the content of $C_{org}$ sharply decreased, while AHCs, on the contrary, increased, that is, the formation of AHCs occurred due to the decomposition of $C_{org}$. At the same time, there were no connections in the distribution of $C_{org}$ and AHC ($r = -0.16$), but a dependence was observed in the distribution of AHCs and PAHs ($r = 0.56$).

The composition of alkanes at station 6841 sharply differed from station 6840 in terms of the content and distribution of homologues (Figure 7). Low molecular weight homologues dominated in their composition, and the L/H ratio increased with the burial depth. The latter testifies to the intensity of autochthonous processes in the sedimentary strata even at a horizon of 22–26 cm. The CPI value at station 6840 (on average 1.8, maximum 2.6) was higher than at station 6841 (on average 1.4, maximum 1.8) may indicate a lesser transformation of high molecular weight homologues directly in the sedimentary sequence. The CPI values usually increase with the depth of burial, since a series of odd more stable homologues increases in the composition of alkanes during the transformation of AHCs [9,22,23]. For comparison, in the Holocene shelf sediments of the Kara Sea, the CPI $C_{22\text{-}33}$ values varied in the range of 2.5–8.1, with an average of 5.2 [24].

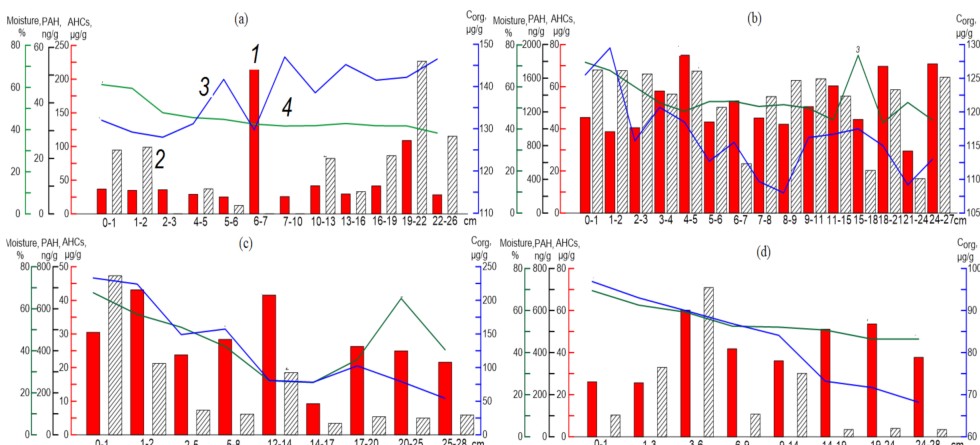

**Figure 6.** Changes in the concentrations of AHCs (1), PAHs (2), $C_{org}$ (3) and moisture (4) precipitation with the depth at stations 6841 (**a**), 6847 (**b**), 6860 (**c**) and 6866 (**d**). The location of the stations is shown in Figure 4.

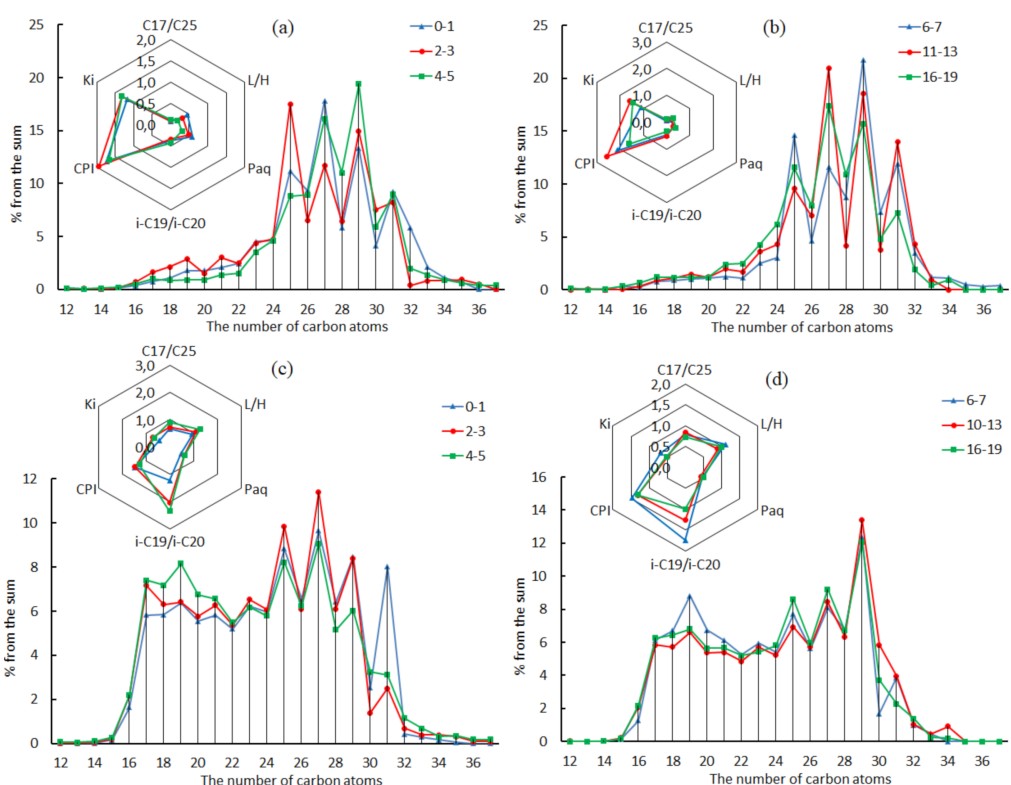

**Figure 7.** Composition of alkanes with burial depth at stations 6840 (**a,b**) and 6841 (**c,d**) and the distribution of the main markers in their composition. The location of the stations is shown in Figure 4.

At the stations in Sturfiord, the composition of PAHs was dominated by 2, 3-ring arenas: naphthalene, 2-methylnaphthalene (27–43% of the total), and phenanthrene (from 22 to 40%) (Figure 8). Naphthalenes are the least stable compounds and should degrade during sedimentation [25], therefore, their rather high content may be due to the formation of sediments directly in the strata. At the same time, at station 6841, the PAH content in the lower core horizon was higher than in the surface one (2633 and 2164 ng/g, respectively).

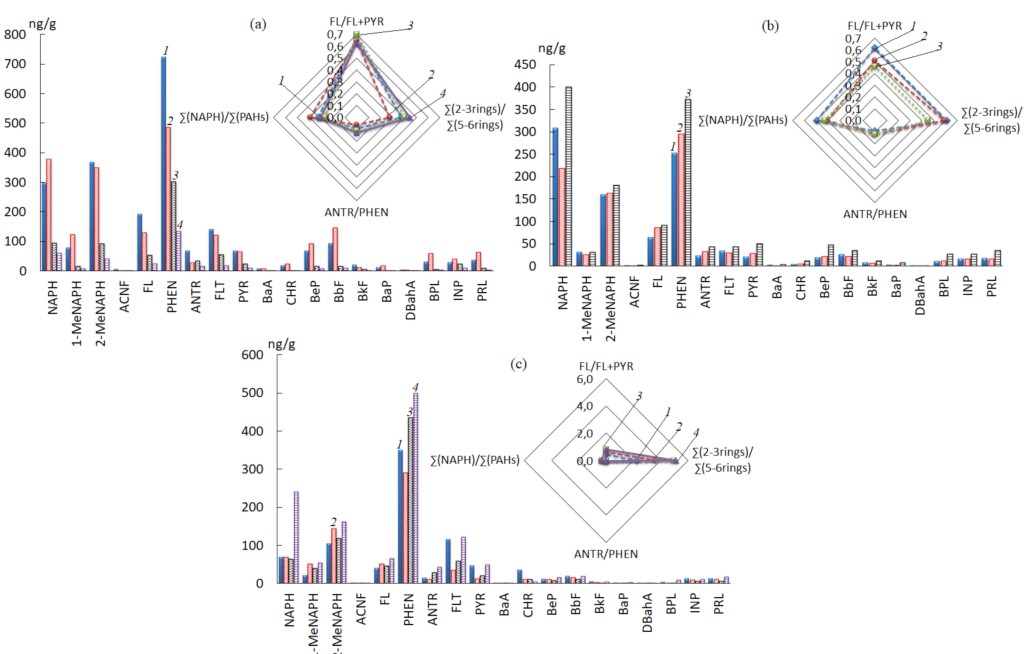

**Figure 8.** Changes in PAH composition with burial depth (**a**) at station 6841: *1*—0-1, *2*—3-4, *3*—5-6, *4*—7-8 cm; (**b**) station 6847: *1*—1–2, *2*—5–6, *3*—24–27 cm; (**c**) station 6864: *1*—0–1 cm, *2*—3–4 cm, *3*—13–16 cm, *4*—24–26 cm. The location of the stations is shown in Figure 4.

Sediments of the Eastern shelf arch. Spitsbergen, represented by strongly bioturbated silty-pelitic silts of a dark yellowish-brown color, oxidized to 8 cm. They also did not show a gradual decrease in the concentrations of $C_{org}$, AHCs and PAHs. The increased concentrations of naphthalene's and phenanthrene are confined not to the surface, but to the 19–22 cm horizon (Figure 6b). In particular, in the sedimentary stratum, there was no dependence in the distribution of AHCs and $C_{org}$ ($r = 0.07$), and, in contrast to $C_{org}$, the AHCs distribution did not depend on the granulometric composition of sediments– $r(AHCs-MOI.). = −0.57$, while $r$ ($C_{org}$-MOI) = 0.58.

It is known that the porosity and moisture content of the sediment characterizes its granulometric composition [26]. Sediments with a high moisture content (up to 90% and more) are formed, as a rule, by a finely dispersed suspension of biogenic origin (for example, fragments of dying planktonic organisms). On the contrary, low moisture values (less than 40%) are characteristic of coarse bottom sediments formed by lithogenic material entering water bodies as a result of erosion of the coastal zone and with slope water runoff.

In the Franz Victoria Trench in the deepest part at station 6860 (depth 3703 m) in the sediment core (Figure 6c), there was a sharp decrease in the content of AHCs (by 4.5 times) and PAHs (by 5.4 times) at the 14–17 cm while the concentration of $C_{org}$ increased almost 2 times (from 1.142 up to 2.107%). At the same time, the composition of the sediment changed, and lenses of dense sandy material appeared in the 13 cm layer, and bioturbation took place from 19 cm. On the shelf in this area (stations 6864 and 6866 with a depth of 594–403 m), an uneven decrease in AHC concentrations was observed in the sediment mass against the background of a decrease in the $C_{org}$ content. At a horizon of 3–6 cm, the AHC content increased more than twofold (up to 60 μg/g) and PAHs (up to 638 ng/g). A similar distribution of organic compounds was observed in the sediment layer at station 6864, where, in the transition from the oxidized to the reduced layer, the amount of naphthalene's increased from 29 to 36% in the PAH composition (Figure 8c).

## 4. Discussion

The data obtained in 2019–2020 show that the anthropogenic input of HCs into bottom sediments were limited by the coastal areas, where their content in the composition of $C_{org}$ increases. In particular, in 2019, with an AHC content of 64 μg/g and PAH 600 ng/g

(Figure 2) in the sandy sediments of the Kaninsky Bank (Moisture 17.4%), their concentration reached an anomalously high value in the $C_{org}$ composition up to –11.7%. Usually, in bottom sediments, the AHC content in the $C_{org}$ composition was less than 0.5%, and the PAH content was < 0.002% (Figure 9). It is believed that the background concentration of AHCs in coarsely dispersed sediments were 10 µg/g, and in finely dispersed sediments—50 µg/g [5,27]. Only in water areas with anthropogenic oil inflows, mud volcanism, and endogenous migration did the AHC concentration increase and exceed 1% in the $C_{org}$ composition. [5,25,27] Therefore, the increase in the AHC value relative to the background concentration, both in terms of dry sediment and in the composition of $C_{org}$, is most likely due to anthropogenic sources.

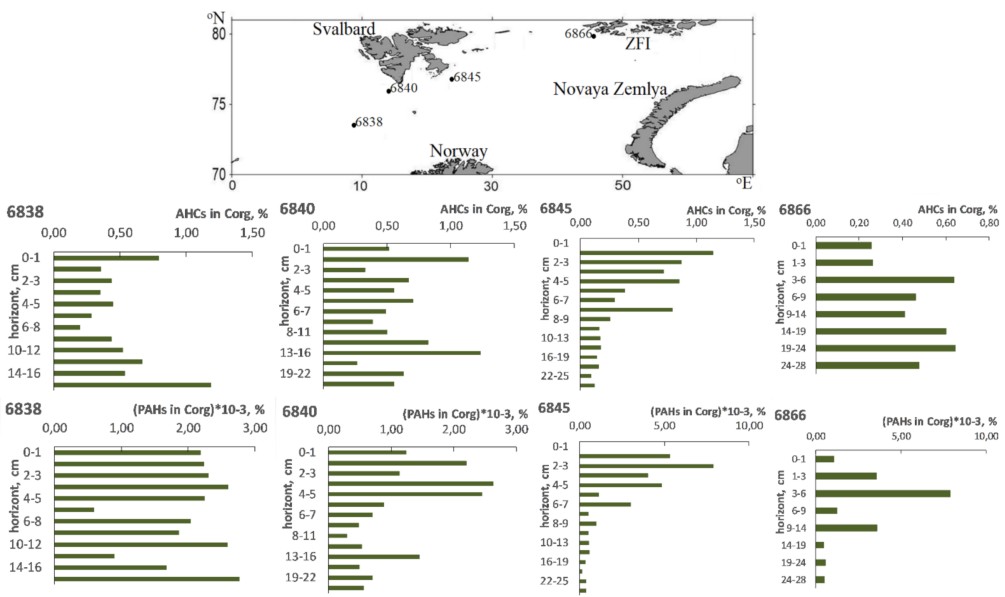

**Figure 9.** Change in the content of AHCs and PAHs in the composition of $C_{org}$ (%) with the depth of burial at individual stations. The location of the stations is shown in Figure 4.

According to our data of 2020, in the coastal sandy sediments (moisture content 11–14%) in the Murmansk region with AHC content of 54–73 µg/g, their proportion in $C_{org}$ varied in a range 3.4–3.6%. However, the composition of alkanes did not correspond to the smooth petroleum distribution of homologues [5,7], since; in the low-molecular-weight region odd homologues of n-$C_{15}$–$C_{19}$ dominated, while in the high-molecular region prevailed $C_{25}$ and $C_{27}$. This is due to the rapid transformation of low molecular weight petroleum alkanes. Even after the diesel fuel spill in Norilsk in May 2020, 2 months after the accident, despite the rather low Arctic temperatures, the composition of alkanes in the surface layer of bottom sediments did not match the composition of the spilled oil product [28].

The concentration of PAHs in sediments of the Murmansk shelf was also high—11,900–13,600 ng/g, but their content in $C_{org}$ was 0.8–0.9% only, and in marine sediments, mostly less than 0.002% (Figure 9). For PAHs, it is more difficult to establish background concentrations, since their values depend on the amount of determined individual polyarenes. In the surface layer of sediments of the Barents Sea ($\sum$22 PAHs), their content varied from 82 to 3076 ng/g with the highest values on the Svalbard shelf [29]. In the southwestern part of the Barents Sea, the PAH content varied from 10 to 1799 ng/g, and in the sediments of the Norwegian fjords they were predominantly of pyrogenic origin [30,31]. According to our 2020 data, the concentration of $\sum$19 PAHs in the surface layer of bottom sediments varied from 2 to 2436 ng/g, which fits into the range of their determined earlier values in the sediments of the Barents Sea (Figure 4). Moreover, the highest concentrations are also found on the Svalbard shelf. Earlier, their increased values were also noted in the carbonaceous sediments of the shelf ([29–35] and others). Consequently, the specificity of

this anomaly is stable. The erosion of carbonaceous deposits in the western part of Svalbard was regarded as the main source of PAHs. High concentrations of pyrogenic PAHs here are due to their natural formation in the sediment mass in low-temperature processes, since they were dominated by low-molecular compounds: naphthalene's and phenanthrene (Figure 8). At the same time, statistically significant differences in the concentrations and composition of PAHs in the sediments sampled in 1991–1998 and 2001–2005 [32,33], as well as in comparison with our data were absent.

In the bottom sediments of the Mohns and Knipovich Ridges, the characteristic structural elements are axial volcanic uplifts (AVP), which are confined to the rift valley [35–39]. Within the AVP, there are rifts/volcanic edifices with hydrothermal activity, and the release of gas/fluid creates hydro acoustic anomalies above these fields [40,41]. In this area, due to the coarsely dispersed composition of the sediments, in the surface layer, we have established rather low concentrations of AHCs, $C_{org}$ and PAHs (Figures 2 and 3). At the same time, the average AHC concentrations in different years of research in the Mohns Ridge area were similar: 10–18 µg/g, with a rather low content of $C_{org}$ in the composition of 0.13–0.28%. These results are consistent with the study of the hydrothermal plume of the Trollveggen field located east of the axial zone of the Mohns Ridge near the Jan Mayen hot spot [42]. The plume of this area was characterized by a moderate concentration of methane and a low concentration of suspended matter near the bottom, while the sediments were characterized by a rather low content of $C_{org}$—0.35% [43]. Nevertheless, with the maximum content at station 6131 (51 µg/g, Figure 2), the proportion of AHCs in the $C_{org}$ composition increased to 1.9%, and in the composition of alkanes in the low-molecular-weight region there was a peak of n-$C_{16}$ (Figure 3a) indicating the microbial nature of AHCs. In addition, in the sandy-silty-pelitic silts at station 6838, located in the area of ancient volcanic edifices, the AHC concentrations increased towards the lower horizon of the core (16–18 cm) to 36 µg/g, 1.2% in the $C_{org}$ composition (Figure 9). At this station, boulder-sized basalt was raised, presumably taken from the side of a volcanic edifice.

In the Barents Sea, the nature of Holocene sediments is mainly marine terrigenous with a noticeable influence of alluvial facies in the coastal part of the shelf and ice-marine in the north of the water area [44]. Most of the sediments studied by us are represented by terrigenous carbonate-free aleurite and silty-pelitic oozes with an admixture of coarse-detrital material. Early diagenesis occurs under conditions of thermodynamic no equilibrium, and bioturbation complicates these processes [45].

On the slope of the Sturfiord trench at a depth of 392 m (station 6842, Figure 4), with the maximum AHC content (186 µg/g), their proportion in the $C_{org}$ composition was also increased—1.18%. In this area, at station 6841, according to hydrophysical data, the most significant fluid flow was established. The gas torch rose above the bottom to a height of over 100 m [15]. The AHC content in the surface layer were only 37 µg/g (0.24% of $C_{org}$). However, the composition of alkanes in sediments at station 6841 sharply differed from their composition at station 6840, first of all, by the content of light homologues (Figure 6). In the northern part of the Medvezhinsky Trench, where there are craters formed as a result of the decomposition of gas hydrates [46,47], an increased AHC content was also found in 2017—up to 44 µg/g, with an increased proportion of low molecular weight homologues (L/H varied in the range 0.84–1.42) [43]. At the same time, during the transition from the oxidized to the reduced layer, the composition of alkanes became more "autochthonous" than in the surface horizon.

In the Franz-Victoria Trough, the change in the HC content in the sediment strata (Figure 9, station 6866) is most likely due to the presence of creep displacements here, which were found during the bathymetric survey.

The abnormal distribution and composition of HCs was established in the sediments of the Perseus Rise in 2017 (at depths of 107–200 m) [43,47]. Here, in the sedimentary strata, when passing from the 0–5 to the 5–10 cm layer, the AHC concentration increased by 53 times, and in the $C_{org}$ composition—by 66 times (from 0.03 to 2.0%). Maximum AHC values at this station in terms of dry sediment (272 µg/g) and in the composition of $C_{org}$

(2.2%) were confined to the 15–20 cm horizon. Such changes in the sedimentary strata can occur during the transformation of seeping oil hydrocarbons [48], since the sediments of this region have a high petroleum and gas generation potential [16].

Fluid flows and their transformation in the surface layer of bottom sediments were considered as the main source of HCs in the study of bottom sediments in the area of the Stockman area [5,6]. The composition of alkanes in sediments had a mixed genesis: autochthonous homologues (n-$C_{16}$–$C_{17}$) dominated in the low-molecular-weight region and petroleum homologues in the high-molecular region; in the composition of PAHs—light polyarens [5]. It was assumed that rather low AHC concentrations in terms of dry weight (in the surface layer 4.4–18.6 µg/g and at a horizon of 10–20 cm—7.8–84.6 µg/g) and in the composition of $C_{org}$ (on average $\leq$ 1%) in this area due to a decrease in the intensity of fluid flows in recent years. It should be borne in mind that the hydrocarbon deposits of the Stockman field are overlain by an impermeable stratum of predominantly clayey rocks [49].

Thus, in the Norwegian-Barents Sea basin, against the background of lateral variability of hydrocarbon molecular markers of bottom sediments, hydrocarbon anomalies of different genesis are distinguished. In the southwestern part of the Barents Sea, anomalies are associated with an anthropogenic component due to the influence of Atlantic waters, coastal runoff, coastal abrasion, aerosol flow and increased shipping. Therefore, in the sediments of the southwestern part of the Barents Sea, PAHs were found to contain compounds indicating air emissions from aluminum smelters when burning coal and wood [34].

Anomalies in the distribution of HCs in bottom sediments on Svalbard shelf, the Medvezhinsky and Sturfiord trenches suggest their natural formation in the sedimentary strata, which determines the specifics of their behavior. As a source of HCs, one can consider their input from the underlying horizons, since they dominate in most samples in the composition of $C_{org}$ (Figure 9). Considering the high petroleum and gas potential of the Barents Sea and the features of the seabed surface (pockmark craters) make this assumption quite reasonable [50–54]. The existence of periods of rapid subsidence, as well as the existence of bituminous rocks, is a reliable indicator of the possible accumulation of a significant amount of HCs.

It is believed that low molecular HCs can move in fluid flows as a separate phase through the pores of sedimentary rocks and leave a geochemical trace in surface sediments due to accumulation, especially in places of gas discharge [55,56]. Low CPI values, indicating a low degree of alkane degradation, as well as the presence of low molecular weight PAHs, can serve as a confirmation of this assumption. Therefore, the sediments of the Barents Sea can be considered as a dynamic generating system that is a function of geological space and time [57].

## 5. Conclusions

The AHC concentration in the surface layer of bottom sediments in 2019 varied in the range of 6–64 µg/g with a maximum in the surface layer of bottom sediments of the Kaninsky Bank of the Barents Sea (11.7% in the composition of $C_{org}$), and in 2020, in the range 3–186 µg/g, with a maximum at Sturfiord on the Svalbard shelf (1.18% of $C_{org}$). A slight predominance of low molecular weight homologues (L/H $\leq$ 1) indicates the intensity of autochthonous processes in bottom sediments.

The PAH concentration in the surface layer of bottom sediments in 2019 varied in the range of 32–9934 ng/g with a maximum on the western shelf of Svalbard, and in 2020, 3–2430 ng/g, with a maximum on the eastern shelf of Svalbard. The PAHs were dominated by phenanthrene and naphthalene's, which is due to their natural formation in the sediment mass in low-temperature processes.

In the sedimentary strata, the lack of correlation in the distribution of HCs with the grain size type of sediments, the content of $C_{org}$, as well as changes in hydrocarbon molecular markers may indicate an endogenous effect in most of the studied areas (in particular, on the shelf of the Svalbard archipelago in Sturfiord and Medvezhinsky

troughs, etc.). The enrichment of the sedimentary section with light alkanes and naphthalene's may be due to outbursts during point discharge of gas fluid from sedimentary rocks of the lower stratigraphic horizons.

In coastal areas on shipping lanes, the organic-geochemical background of bottom sediments is formed due to sedimentation processes, which leads to an increase in HC concentrations in the surface layer and in the composition of $C_{org}$ (on the Kaninsky Bank up to 64 µg/g for AHCs and 600 ng/g for PAHs). However, the anthropogenic input into the bottom sediments of the Barents Sea is of subordinate importance in comparison with their natural input into fluid flows.

**Supplementary Materials:** The following are available online at https://www.mdpi.com/article/10.3390/fluids6120456/s1, Table S1: Characteristics of the surface layer of bottom sediments in the Norwegian and Barents Seas in different years of research, Table S2: Concentrations of organic compounds in bottom sediments at individual stations and distribution of markers in the composition of PAHs.

**Author Contributions:** I.A.N. collected samples in the 75 and 80 of the R/V Akademik Mstislav Keldysh and isolated hydrocarbons on board the vessel and interpreted the data obtained; A.V.K. determined hydro-carbons in laboratory conditions, analyzed alkanes, made a figure for the paper, All authors have read and agreed to the published version of the manuscript.

**Funding:** The expeditions were carried out within the framework of the state assignment of the Ministry of Education and Science of Russia (№ 0128-2021-0015), geochemical research and generalization of materials with the financial support of the Russian Science Foundation (project No. 19-17-00234).

**Institutional Review Board Statement:** Not applicable.

**Informed Consent Statement:** Not applicable.

**Data Availability Statement:** The data presented in this study are openly available in FigShare at.

**Acknowledgments:** The authors are grateful to Gordeev V.V. their valuable comments and advice when discussing the results; Halikov I., Popova M., Solomatina A., Chernov V. for assistance in conducting the analyses r.

**Conflicts of Interest:** The authors declare no conflict of interest.

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
