# Peer review of "Features of the Hydrocarbon Distribution in the Bottom Sediments of the Norwegian and Barents Seas"

_fluids, doi:10.3390/fluids6120456_

Round 1

Reviewer 1 Report

Review of the manuscript

Features of the Hydrocarbon Distribution in the Bottom Sedi-ments of the Norwegian and Barents Seas

By

 I.A. Nemirovskaya  and A.V. Khramtsova

Manucript deals with a study of hydrocarbons when found with n the bottom sediments found along the Norwegian-Barents Sea basin: Mohns Ridge, shelf Svalbard archipelago, Sturfiord, Medvezhinsky trench, central part of the Barents Sea, Novaya Zemlya shelf, Franz Victoria trough.

The origin of the contribution refers to 2019 and 2020 year in the bottom sediments of the Norwegian and Barents Seas.

From methodological point of view, manuscript presents a very simple procedure to collect the sediments, prepare them for the analyses and spectral analysis.

Reference list looks kind of relevant and up to date, implying pretty good overview of the are of interest by the authors.

Below I attach several comments to be adressed by the authors:

The results of the study of hydrocarbons (HCs): aliphatic (AHCs) and polycyclic aromatic hydrocarbons (PAHs) in bottom sediments (2019 and 2020, cruises 75 and 80 of the R/V Akademik Mstislav Keldysh) in the Norwegian-Barents Sea basin: Mohns Ridge, shelf Svalbard archipelago, Sturfiord, Medvezhinsky trench, central part of the Barents Sea, Novaya Zemlya shelf, Franz Victoria trough.

This sentence is far from being clear to the reader. Besides, verb is missed from the sentence which make sit non sense. Authors should improve the meaning to be said and the clearness of the sentence. I warmly suggest reorganization.

Figure 2. Legend needed urgently. Difference between red bars and yellow line should be exactly explained. Whats the meaning of numbers? For example the be located most to the west is denoted by numbers 51 and 7. In the same time it is linked by the red line to number id 6134. Sorry, but assuming that someone will understand your thoughts automatically is non realistic. Please give effort and make Figure clear to the reader in away of both, presence and interpretation.

Figure 3. Axis values and variables should be clerly presented.

Figure 4. What is represented by hatch bar? What is represented by red bar? What is represented by purple circuit? What are values around bars?

Figure 8. Please make variables and values bigger  so one can read them easily.

What is the scientific contribution of the proposed work. I would appreciate to be clarified and added to discussion section.

Why the manuscript starts with 2017 and 2018 data sets if only 2019 and 2020 have been analysed?

What is the practical contribution of the conclusions arose from this study?

Reviewer 2 Report

On the whole, the work is interesting.

However, I have some comments on this manuscript that need to be improved.

  1. Some parts of the manuscript need to be restructured.
    • The text of the Results section and should be divided into subsections. The description is not clear in the current form of the text.
    • From the Results section, all interpretations of results should be moved to the Discussion section.
    • No information in the text about the geology of the study area. This should be described in a separate section containing a map and a cross-sections.
    • The Introduction section should be shortened and more general. Detailed results of previous studies should be transferred to the Discussion section and discussed there in the context of the current results.
  2. When describing the results, authors should refer to specific areas of the Barents Sea, not just station numbers. And these geographical names and names of geological structures should be marked on the map (all mentioned in the text).
  3. The research methodology needs to be supplemented because:
    • no information on lithological and grain size analysis methods
    • no information about the methods of moisture analysis
    • no information about the methods of Eh analysis

Results of these analysis are described in Results, Discussion and Conclusion sections.

    • no information on the number of samples
    • no information on the depth of sampling
    • no information on the methodology of statistical analyzes. Were they made? They should be done since there is information about averages and correlations in the text.
  1. The author writes that "the purpose of the research is to obtain new data (2019 and 2020) on the spatial distribution ...". Therefore, I recommend making maps that present such spatial distribution.
  2. The figures are not clear. They need to be corrected and make with more accuracy.
  3. Figure 2 – there are no green column

Round 2

Reviewer 1 Report

No comments.

Reviewer 2 Report

The manuscript has been revised and can be accepted in its present form for publication.